# TARGET-AWARE NORMALIZED DISTILLATION: A PRINCIPLED FRAMEWORK FOR ROBUST KNOWLEDGE TRANSFER

## ABSTRACT

*Knowledge Distillation* (*KD*) has become a cornerstone for model compression, semi-supervised learning, and self-training. Despite its success, the standard KL-based objective suffers from a structural flaw: it *couples* supervision on target and non-target classes. This coupling links the estimation of target probability mass to the loss on non-target probabilities, thereby amplifying mass mismatch and destabilizing optimization under noise or teacher miscalibration. To address this issue, we propose *Target-Aware Normalized Distillation* (*TAND*), a principled framework that explicitly decouples and normalizes distillation signals. TAND combines *Normalized KD* (*NKD*), which aligns the normalized non-target distributions of student and teacher, with *Target-Aware Distillation* (*TAD*), which assigns independent weights to target and non-target terms. This explicit decoupling breaks the hidden dependency in KD, stabilizes gradient dynamics, and offers direct control over supervision strength. We theoretically prove that TAND reduces gradient variance, explaining its robustness, and empirically validate its effectiveness on *distantly-supervised NER*, *noisy-label learning*, and *transfer gap* tasks. Across all settings, TAND consistently outperforms KL-based KD baselines, demonstrating strong robustness to noise across different noise levels and model architectures.

## 1 INTRODUCTION

*Knowledge Distillation* (*KD*) has emerged as a fundamental paradigm in modern machine learning. Initially introduced for *model compression*, where a large *teacher* model transfers knowledge to a smaller *student* (Hinton et al., 2015), it has since evolved into a general framework underpinning *self-training* and *semi-supervised learning*, thereby enabling the effective utilization of unlabeled data. Beyond efficiency, KD is also regarded as a *noise-tolerant* mechanism: teacher posteriors provide smoothed supervision that mitigates overfitting to noisy or imperfect labels (Li et al., 2020; Han et al., 2018). This dual promise of *scalability* and *robustness* has rendered KD a cornerstone technique across vision, natural language processing, and speech recognition.

Nevertheless, the *canonical* KD objective—namely, the Kullback–Leibler (KL) divergence between teacher and student distributions—exhibits a fundamental structural limitation. Specifically, it implicitly *couples* the supervision of the *target* and *non-target* classes. While this coupling is benign under well-calibrated teacher predictions, it becomes detrimental under label noise or miscalibration: non-target probabilities induce spurious gradients scaled by unreliable target confidence, leading to unstable optimization and degraded generalization. As a result, KL-based KD often proves fragile in realistic noisy-label regimes.

Substantial effort has been devoted to improving KD. Response-based methods align student and teacher posteriors, feature- and attention-based methods transfer intermediate representations (Romero et al., 2015; Zagoruyko & Komodakis, 2017), and relation-based methods distill higher-order structural knowledge (Xu et al., 2020). Self-distillation eliminates the explicit teacher, iteratively reusing the student across generations (Zhang et al., 2019). Although these approaches enrich the form of knowledge transferred, they typically retain the same KL-based objective and thus inherit its structural deficiencies.

Another line of research has focused on robustness under noisy or weak supervision. Representative approaches include Noisy Student (Xie et al., 2020) and Mean Teacher (Tarvainen & Valpola, 2017), which stabilize pseudo-labeling via exponential moving average (EMA) teachers and strong augmentation; confidence-based filtering and temperature scaling, which suppress unreliable predictions (Guo et al., 2017; Li et al., 2020); and methods in noisy-label learning, such as small-loss selection (Han et al., 2018; Yu et al., 2019), robust loss functions (Zhang & Sabuncu, 2018; Wang et al., 2019), or semi-supervised perspectives that down-weight low-confidence samples (Li et al., 2020). Although effective in mitigating certain types of noise, these strategies do not fundamentally resolve the *structural pathology* of KL-based KD: non-target supervision remains scaled by target confidence, and mismatches in non-target probability mass persist.

Other related techniques, such as label smoothing (Szegedy et al., 2016) and calibration adjustments (Guo et al., 2017), redistribute probability mass or rescale logits, but neither achieve explicit decoupling between target and non-target supervision nor eliminate the mass mismatch between teacher and student distributions. Similarly, in distantly supervised named entity recognition (DS-NER), prior work has improved label quality through boundary-aware tagging, partial CRFs, or token selection (Shang et al., 2018; Liang et al., 2020; Zhang et al., 2021), yet the KD objective itself remains unmodified.

In this paper, we revisit the KD objective and introduce a principled reformulation specifically designed for noisy or weakly supervised scenarios. Unlike existing approaches that rely on auxiliary heuristics such as data filtering or probability rescaling, our method directly redefines the distillation loss. By algebraically decomposing KL divergence into a binary target-vs-rest term and a conditional non-target term, we reveal the hidden coupling that destabilizes gradient dynamics. Building upon this insight, we propose *Target-Aware Normalized Distillation* (*TAND*), a task-agnostic framework that: *(i) normalizes* non-target distributions to eliminate mass mismatch, and *(ii) decouples* target and non-target supervision through explicit weighting. This formulation renders supervision transparent, tunable, and robust to teacher miscalibration. Moreover, TAND is orthogonal to existing denoising and semi-supervised strategies and can be seamlessly integrated with diverse model architectures, temperatures, and teacher constructions.

The contributions of this work are threefold: *(i)* We identify and formalize a structural pathology in KL-based KD, showing that non-target supervision is implicitly scaled by teacher target confidence and that mismatched non-target mass introduces destabilizing residual gradients. *(ii)* We propose *Target-Aware Normalized Distillation (TAND)*, which integrates *Normalized KD (NKD)* to remove mass mismatch with *Target-Aware Distillation (TAD)* to decouple supervision via explicit weighting. *(iii)* We provide both theoretical analysis and empirical validation, demonstrating that TAND reduces gradient variance and consistently improves robustness across regimes reflecting real-world challenges, including *distantly-supervised NER*, *noisy-label learning*, and *transfer gap*. Comprehensive experiments on NLP and vision tasks confirm that TAND surpasses KL-based KD and strong noisy-label baselines, establishing its robustness to noise, distributional perturbations, and model scaling.

## 2  KNOWLEDGE DISTILLATION: FORMULATION AND LIMITATIONS

We begin by revisiting the standard formulation of *Knowledge Distillation* (*KD*), establishing notation and analyzing its inherent structural deficiencies. In particular, by decomposing the canonical KL divergence objective into *target* and *non-target* components, we uncover two fundamental pathologies—*magnitude coupling* and *mass mismatch*—that compromise stability under noisy supervision and teacher miscalibration. This analysis provides the theoretical motivation for the framework introduced in Section 3.

Let $\mathcal{X}$ denote the input space and $\mathcal{Y} = \{1, \dots, C\}$ the label space with $C$ classes. For any input $x \in \mathcal{X}$, the *teacher* and *student* networks output logits $z^t, z^s \in \mathbb{R}^C$. With a temperature parameter $\tau > 0$, the softened class probabilities are defined as

$$p_c^t = \frac{\exp(z_c^t/\tau)}{\sum_{k=1}^{C} \exp(z_k^t/\tau)}, \qquad p_c^s = \frac{\exp(z_c^s/\tau)}{\sum_{k=1}^{C} \exp(z_k^s/\tau)}, \quad c \in \{1, \dots, C\}.$$

The canonical KD objective (Hinton et al., 2015) minimizes the Kullback–Leibler (KL) divergence between teacher and student distributions:

$$L_{\text{KD}} = \text{KL}(p^t \,\|\, p^s) = \sum_{c=1}^{C} p_c^t \cdot \log \frac{p_c^t}{p_c^s} = \sum_{c=1}^{C} -p_c^t \cdot \log p_c^s + \sum_{c=1}^{C} p_c^t \cdot \log p_c^t. \tag{1}$$

Let $t \in \mathcal{Y}$ denote the ground-truth class. We partition the teacher distribution into the *target mass* $p_t^t$ and the *non-target mass* $1 - p_t^t$, and define the normalized non-target distributions as

$$\rho_c^t = \frac{p_c^t}{1 - p_t^t}, \quad \rho_c^s = \frac{p_c^s}{1 - p_t^s}, \quad c \neq t, \tag{2}$$

with shorthand notations $\rho^t := (\rho_c^t)_{c \neq t}$ and $\rho^s := (\rho_c^s)_{c \neq t}$.

**Theorem 2.1** (Decomposition). *The KL divergence in Eq. (1) admits the decomposition*

$$L_{\text{KD}} = \underbrace{\text{KL}\big([p_t^t, 1 - p_t^t] \,\|\, [p_t^s, 1 - p_t^s]\big)}_{\textit{Target-Class KD (TCKD)}} + (1 - p_t^t) \cdot \underbrace{\text{KL}(\rho^t \,\|\, \rho^s)}_{\textit{Non-Target Class KD (NCKD)}}, \tag{3}$$

*where*

$$L_{\text{TCKD}} := \text{KL}\big([p_t^t, 1 - p_t^t] \,\|\, [p_t^s, 1 - p_t^s]\big), \tag{4}$$

$$L_{\text{NCKD}} := \text{KL}(\rho^t \,\|\, \rho^s). \tag{5}$$

Eq. (3) in Theorem 2.1, whose proof is given in Appendix A.1, highlights a latent *coupling effect* in KL-based KD:

$$L_{\text{KD}} = \text{TCKD (target vs. rest)} + m^t \cdot \text{NCKD (normalized non-target)}, \tag{6}$$

where $m^t := 1 - p_t^t$ denotes the teacher's non-target mass.

Two fundamental deficiencies follow from this structure:

*(i)* **Magnitude coupling.** The contribution of NCKD is scaled by $m^t$, which is an empirical estimate of the true non-target mass. Since $m^t$ is subject to fluctuations caused by label noise, weak supervision, or calibration errors, the effective strength of non-target supervision is rendered unstable. This multiplicative dependence directly inflates gradient variance, thereby compromising convergence stability (see Section 3.4).

*(ii)* **Mass mismatch.** Although NCKD enforces alignment of the *shape* of non-target distributions, its weight in the overall loss is determined by $m^t$. If $m^t$ deviates substantially from the true non-target mass—e.g., when underestimated or nearly vanishing—the contribution of NCKD becomes insufficient. Consequently, target mass mismatch may lead to limited supervision on alignment of $\rho^s$ and $\rho^t$, thereby hindering the knowledge transfer on the non-target classes.

These observations indicate that a robust distillation objective should satisfy the following desiderata:

  *(i) Decoupling*: target and non-target supervision should be separated, removing multiplicative dependence on $m^t$;

  *(ii) Normalization*: non-target distributions should be aligned independently of mass estimation accuracy;

  *(iii) Explicit control*: the relative strengths of supervision should be tunable via weighting parameters, ensuring robustness to noise, oscillation, and miscalibration.

In summary, although KL-based KD decomposes naturally into target and non-target components, its hidden *coupling effect*—manifested in magnitude coupling and mass mismatch—introduces unnecessary variance and bias. These deficiencies account for the observed fragility of standard KD in practice and directly motivate the development of *Target-Aware Normalized Distillation (TAND)*, presented in Section 3.

## 3 METHODOLOGY: TARGET-AWARE NORMALIZED DISTILLATION (TAND)

The preceding analysis identified two fundamental deficiencies in the KL-based KD objective: *(i)* *mass mismatch*, whereby the teacher's non-target probability mass can be underestimated, thereby diminishing the effective contribution of NCKD to non-target alignment; and *(ii) magnitude coupling*, whereby the non-target supervision is scaled by the teacher's non-target confidence $(1 - p_t^t)$, introducing instability through fluctuations in the target mass estimate $p_t^t$. These limitations collectively explain the empirical fragility of conventional KD under noisy supervision, calibration drift, or distributional shift.

To address these issues, we introduce *Target-Aware Normalized Distillation* (*TAND*), a principled framework that explicitly *decouples* and *normalizes* the supervision signals. The overarching goal of TAND is to suppress gradient variance, mitigate error propagation from teacher miscalibration, and provide explicit and tunable control over supervision strength. The framework comprises two complementary components: *(i) Normalized Knowledge Distillation* (*NKD*), which eliminates the adverse effect of mass mismatch on aligning the *shape* of the non-target distributions, and *(ii) Target-Aware Distillation* (*TAD*), which resolves magnitude coupling by assigning explicit weights to target and non-target components, thereby removing implicit dependence on the teacher's mass.

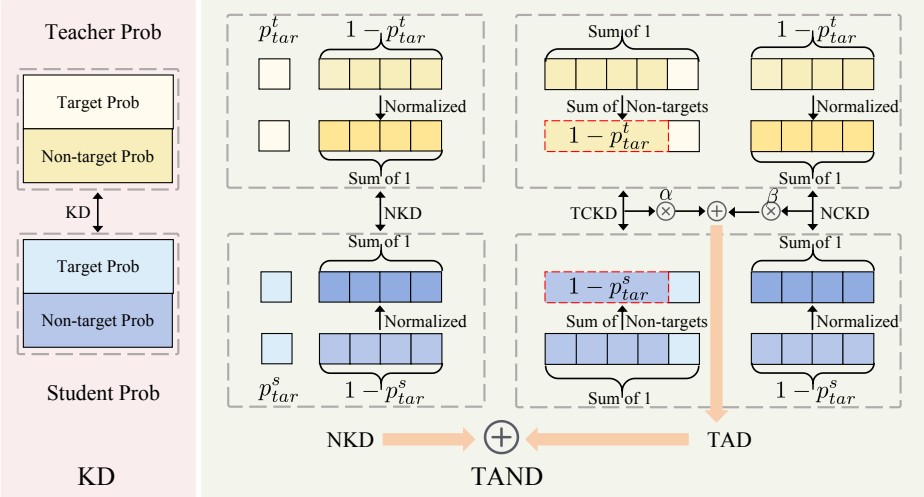

Figure 1: Illustration of our proposed Target-Aware Normalized Distillation (TAND) and traditional KD. In TAND, NKD enforces alignment of normalized non-target distributions, while TAD decouples target and non-target terms through explicit weighting with parameters $\alpha$ and $\beta$.

### 3.1 NORMALIZED KNOWLEDGE DISTILLATION (NKD)

In standard KD, the student is compelled to match both the absolute target mass and the relative structure of non-target probabilities. However, since the teacher's non-target mass $m^t$ often deviates from the ground-truth mass, this discrepancy biases the ratio of the NCKD component in KD Eq. (6), which is intended to capture the relative structure of the non-target class probabilities.

To overcome this issue, we normalize the non-target probabilities into distributions $\rho^t$ and $\rho^s$ as in Eq. (2). A balancing coefficient $\gamma > 0$ is introduced to regulate the relative importance of target and non-target terms. The NKD loss is defined as

$$L_{\text{NKD}} = -p_t^t \log p_t^s - \gamma \sum_{k \neq t} \rho_k^t \log \rho_k^s. \tag{7}$$

By construction, NKD aligns the *relative structure* of the non-target distribution by removing the influence of absolute mass, while simultaneously preserving classification performance by enhancing the target class mass of the student network. This formulation ensures that the student captures the structural regularities of the teacher's non-target outputs, enhancing stability and robustness under mass mismatch.

## 3.2 TARGET-AWARE DISTILLATION (TAD)

Normalization alone is insufficient to resolve magnitude coupling. In canonical KD, the weight of the non-target term remains scaled by $m^t = 1 - p_t^t$, compelling the student to replicate fluctuations in the teacher's non-target confidence.

To eliminate this dependency, we propose *Target-Aware Distillation* (*TAD*), which introduces explicit weights for target and non-target terms:

$$L_{\text{TAD}} = \alpha \cdot L_{\text{TCKD}} + \beta \cdot L_{\text{NCKD}}, \tag{8}$$

where $\alpha, \beta > 0$ are tunable hyperparameters, and

$$L_{\text{TCKD}} = \text{KL}\big([p_t^t, 1 - p_t^t] \,\|\, [p_t^s, 1 - p_t^s]\big),$$
$$L_{\text{NCKD}} = \text{KL}(\rho^t \,\|\, \rho^s).$$

Unlike the canonical objective KD, TAD regulates the strength of non-target supervision through $\beta$, rather than relying on the random variable $m^t$. This explicit control enables fine-grained adjustment: larger $\alpha$ emphasizes distillation of the target class mass, while larger $\beta$ enforces structural consistency among non-target categories. Thus, TAD transforms implicit coupling into a tunable mechanism, enhancing robustness to noisy or miscalibrated supervision.

## 3.3 UNIFIED OBJECTIVE: TARGET-AWARE NORMALIZED DISTILLATION (TAND)

The final TAND objective integrates NKD and TAD into a single formulation:

$$L_{\text{TAND}} = L_{\text{NKD}} + L_{\text{TAD}}. \tag{9}$$

This construction simultaneously addresses the two structural deficiencies: *(i)* NKD resolves non-target mass mismatch, ensuring stable gradients when aligning non-target distributions; and *(ii)* TAD decouples target and non-target supervision, preventing fluctuations in $m^t$ from corrupting non-target gradients.

Taken together, NKD and TAD enable TAND to *remove hidden coupling*, *resist mass mismatch*, and *stabilize gradient dynamics*. As will be shown in Section 3.4, these properties provide a theoretical explanation for the empirical robustness of TAND across diverse regimes, including noisy or weak supervision, online teacher refresh, fine-grained classification, and distributional shift.

## 3.4 VARIANCE REDUCTION ACROSS TASKS VIA TAND

As discussed in Section 3, standard KD inherently couples target and non-target supervision through the non-target mass $m^t = 1 - p_t^t$. This coupling introduces the stochastic variability of $m^t$ to the training process, leading to unstable gradients during optimization, especially under noisy or miscalibrated teachers. In contrast, TAND replaces this implicit dependence with explicit coefficients $(\alpha, \beta, \gamma)$, transforming $m^t$ from a random multiplicative factor into a controllable parameter. This reparameterization suppresses variance in gradient norms, leading to more stable optimization. The following theorem formalizes this reduction.

**Theorem 3.1** (Variance reduction). *Let $z^s$ be the logits of the student model. Assume that across training steps the teacher target mass $p_t^t$ exhibits nonzero variance $\text{Var}[p_t^t] > 0$ due to miscalibration or label noise. Then, relative to KD,*

$$\text{Var}\big[\|\nabla_{z^s} L_{\text{TAND}}\|_2\big] \;\leq\; \text{Var}\big[\|\nabla_{z^s} L_{\text{KD}}\|_2\big] \;-\; \kappa \,\text{Var}[p_t^t] \;+\; \underbrace{\mathbb{E}\big[o((p_t^t - \mathbb{E}p_t^t)^2)\big]}_{\text{higher-order terms}},$$

*for some $\kappa > 0$ depending on $(\alpha, \beta, \gamma)$.*

Theorem 3.1, whose proof is given in Appendix A.2, highlights that TAND reduces gradient variance proportionally to the variability of $p_t^t$. As seen in Eq. (6), in KD, KD couples fluctuations in teacher confidence with the loss on non-target classes, amplifying instability and variability. TAND breaks this multiplicative link and replaces it with tunable coefficients, enabling explicit control over gradient contributions and dampening volatility. Theorem 3.1 theoretically reveals the difference in the stability of the two distillation losses, in terms of the norm of the gradient with respect to the logits. We now analyze three representative scenarios to illustrate how TAND consistently reduces variance relative to KD.

**DS-NER.** In distantly supervised NER, annotation noise arises from imperfect label alignment between text spans and knowledge bases. This typically causes inconsistent confidence estimates $p_t^t$ from the teacher, leading to large variance in the non-target mass $m^t$. Under KD, such oscillations directly rescale non-target gradients, forcing the student to mimic unstable supervision and resulting in erratic optimization. By decoupling and normalizing supervision, TAND reduces sensitivity to these fluctuations: the normalized non-target loss focuses only on distributional shape, while explicit weights $(\alpha, \beta)$ ensure that noisy target estimates do not propagate variance into non-target alignment.

**Noisy-Label Learning.** In the noisy-label setting, teacher predictions are often biased by mislabeled or out-of-distribution samples. This induces high variance in $p_t^t$ since the teacher alternates between overconfident but incorrect predictions for noisy labeled samples and uncertain soft outputs for correctly labeled samples. KD amplifies this effect, as the non-target gradient magnitude is stochastically scaled by $m^t$. TAND mitigates this instability by normalizing the non-target distribution and fixing the relative importance of target versus non-target supervision through $(\alpha, \beta, \gamma)$. As a result, variance in gradient norms is significantly reduced, enabling the student to maintain stable learning trajectories even when the input distribution is contaminated with severe noise.

**Transfer Gap.** In the transfer gap scenario, the teacher is trained in a human-labeled domain, but its probability outputs are biased when applied to the machine-labeled domain. Here, variance arises not only from random noise but also from systematic miscalibration, i.e., consistent under- or over-estimation of $p_t^t$ across classes. KD inherits these biases, with the non-target supervision scaled by mismatched $m^t$, thereby distorting the student's gradient updates. TAND, by explicitly decoupling and normalizing supervision, prevents such bias from being amplified. The fixed weighting of TAD ensures that miscalibrated $m^t$ does not destabilize non-target alignment, while NKD enforces structural consistency across normalized non-target distributions. Hence, TAND reduces both stochastic and systematic variance, narrowing the performance gap between teacher and student across domains.

## 4 EXPERIMENTS

Theorem 3.1 and its instantiations across DS-NER, noisy-label learning, and transfer gap collectively demonstrate that TAND suppresses gradient variance under diverse forms of supervision noise. This variance reduction constitutes the central mechanism underlying TAND's robustness. To empirically validate these theoretical insights, we now turn to systematic experiments on the same three representative tasks—*DS-NER*, *noisy-label learning*, and the *transfer gap*. Together, these settings capture the major challenges of weak supervision, noisy or corrupted labels, and distributional shift.

Our experimental evaluation is organized around three main objectives: *(i)* to examine whether TAND consistently improves student performance across tasks when used in place of the canonical KD loss; *(ii)* to assess the contribution of each individual component of TAND (NKD and TAD) to the overall effectiveness of the framework; and *(iii)* to analyze the sensitivity of TAND to its hyperparameters and extract insights regarding its robustness and stability.

### 4.1 EXPERIMENTAL SETUP

#### 4.1.1 TASKS AND DATASETS

**DS-NER.** DS-NER is a canonical setting where weak or noisy annotations are inevitable. The task requires identifying named entities from text with imperfectly aligned labels. We evaluate on standard benchmarks including `CoNLL03` (Sang & De Meulder, 2003), `OntoNotes5.0` (Weischedel et al., 2013), `Webpage` (Ratinov & Roth, 2009), `Wikigold` (Balasuriya et al., 2009), and `Twitter` (Godin et al., 2015). Performance is reported using precision (P), recall (R), and F1 score (F1). This task stresses the robustness of distillation to annotation noise and domain-specific distributional artifacts.

**Noisy-Label Learning.** In this setting, we adopt the data-free KD paradigm, where a powerful pre-trained teacher is available, but clean training data is inaccessible. Instead, only noisy datasets are available, which we construct to simulate real-world data contamination. The dataset consists

of three subsets: *(i)* a *clean set*, sampled from CIFAR-100 (Krizhevsky et al., 2009), preserving $(1-\rho_1) \times 50,000$ original labels, where $\rho_1 \in (0,1)$ denotes the noisy sample ratio; *(ii)* a *closed noisy set*, constructed by randomly flipping the true labels of $\rho_1(1-\rho_2) \times 50,000$ CIFAR-100 samples to incorrect labels within CIFAR-100, where $1-\rho_2 \in (0,1)$ is the proportion of closed noisy samples among all noisy samples; and *(iii)* an *open noisy set*, where $\rho_1\rho_2 \times 50,000$ images from ImageNet are randomly mapped into CIFAR-100 classes. We evaluate on $10,000$ clean CIFAR-100 test samples. This setting directly tests whether TAND can maintain meaningful supervision under severe noise and various types of noise.

**Transfer Gap.** In classical KD, the teacher is trained on human-annotated data, while the student is optimized to mimic the teacher's soft predictions. This creates a *transfer gap* since the teacher's output distributions generally differ from the ground-truth label distributions, which can lead to biased or suboptimal student learning. We mitigate this phenomenon on CIFAR-100, a challenging multi-class dataset where teachers' outputs are often over-confident or under-confident on certain samples, causing these errors to propagate in the student. This experimental setup directly evaluates TAND's ability to address distributional misalignment between teacher and ground-truth domains.

### 4.1.2 BASELINES

**DS-NER.** We compare against fully supervised NER models (RoBERTa-base (Liu et al., 2019), BiLSTM-CRF (Ma & Hovy, 2016)) to establish an upper bound, and several DS-NER methods that address weak labels: AutoNER (Shang et al., 2018) (boundary-aware tagging), Co-teaching+ (Yu et al., 2019) (small-loss selection), NegSampling (Li et al., 2020), and SCDL (Zhang et al., 2021) (self-distillation with confidence denoising). We construct SCDL+TAND by replacing SCDL's KD loss with our TAND objective.

**Noisy-Label Learning.** We evaluate against MODUL (Tang et al., 2024), which combines KD on clean/closed noisy sets with self-supervised learning on open noisy data. Our variant MODUL+TAND replaces MODUL's KD loss with TAND, thereby testing whether explicit decoupling further enhances robustness.

**Transfer Gap.** We compare with IPWD (Niu et al., 2022), which reweights KD loss to reduce bias from teacher predictions. We introduce IPWD+TAND, replacing its KD loss with TAND, to test whether variance reduction directly mitigates transfer gap effects.

### 4.1.3 IMPLEMENTATION DETAILS

**DS-NER.** We adopt RoBERTa and DistilRoBERTa as teachers, with students sharing the same architecture. Following Zhang et al. (2021), we set EMA parameters $\{0.9, 0.99, 0.995, 0.998\}$ and threshold $\delta = 0.9$. Learning rate is $1e^{-5}$, trained for 50 epochs. Pre-training spans 3, 1, 12, and 6 epochs on CoNLL03, OntoNotes5.0, Webpage, and Twitter, respectively. Batch sizes are 8 (CoNLL03, Webpage, Twitter) and 16 (OntoNotes5.0). Unless otherwise specified, we set $\alpha = 1.0$, $\beta = 2.0$ (Eq. (8)) and $\gamma = 1.5$ (Eq. (7)).

**Noisy-Label Learning.** We adopt ResNet34–ResNet18 and VGG16–VGG13 as the network architecture of teacher–student pairs. For MODUL+TAND, we adopt the warm-up training scheme, where the KL-divergence was first adopted in the algorithm to warm-up the student network during the first 50 epochs, and then we use TAND with the parameters $\alpha = 2.0$, $\beta = 0.5$, and $\gamma = 0.5$, as the training objective in the following. We train networks with a batch size of 256. Other experimental details are the same as MODUL.

**Transfer Gap.** We investigate a variety of student-teacher architecture pairs. For IPWD+TAND, we adopt the warm-up training scheme, where KL-divergence was first adopted in the algorithm to warm-up the student network during the first 50 epochs, and then we adopt TAND with the parameters $\alpha = 2.0$, $\beta = 0.5$, and $\gamma = 0.5$, instead of KL, as the training objective in the following. Other experimental details are the same as IPWD in Niu et al. (2022). Further details are provided in Appendix B.

### 4.2 MAIN RESULTS

**DS-NER.** Table 1 reports results across five datasets measured by precision (P), recall (R), and F1 scores. SCDL+TAND consistently outperforms its base model SCDL, with F1 gains of $+1.51\%$

(Wikigold), +1.23% (Twitter), +2.16% (Webpage), and +0.27% (CoNLL03). These improvements highlight that TAND's explicit decoupling allows more stable supervision in the presence of weak labels. Compared to AutoNER and Co-teaching+, our method achieves significantly higher precision and F1, demonstrating resilience to annotation noise. Overall, results show that TAND provides a principled improvement over both denoising strategies and self-distillation frameworks.

Table 1: Experimental results of the compared methods.

| Methods | Webpage | | | Wikigold | | | Twitter | | | CoNLL03 | | | OntoNotes 5.0 | | |
|---|---|---|---|---|---|---|---|---|---|---|---|---|---|---|---|
| | P | R | F1 | P | R | F1 | P | R | F1 | P | R | F1 | P | R | F1 |
| BILSTM-CRF * | 50.07 | 54.76 | 52.34 | 55.40 | 54.30 | 54.90 | 60.01 | 46.16 | 52.18 | 91.35 | 91.06 | 91.21 | 85.99 | 86.36 | 86.17 |
| RoBERTa * | 66.29 | 79.73 | 72.39 | 85.33 | 87.56 | 86.43 | 51.76 | 52.63 | 52.19 | 89.14 | 91.10 | 90.11 | 84.59 | 87.88 | 86.20 |
| AutoNER | 48.82 | 54.23 | 51.39 | 43.54 | 52.35 | 47.54 | 43.26 | 18.69 | 26.10 | 75.21 | 60.40 | 67.00 | 64.63 | 69.95 | 67.18 |
| Co-teaching+ | 61.65 | 55.41 | 58.36 | 55.23 | 49.26 | 52.08 | 51.67 | 42.66 | 46.73 | 86.04 | 68.74 | 76.42 | 66.63 | 69.32 | 67.95 |
| SCDL | 68.71 | **68.24** | 68.47 | 62.25 | 66.12 | 64.13 | **59.87** | 44.57 | 51.09 | **87.96** | 79.82 | 83.69 | **67.49** | 69.77 | 68.61 |
| SCDL+TAND | **70.22** | 65.54 | **71.32** | **63.95** | **67.44** | **65.65** | 57.96 | **47.68** | **52.32** | 86.56 | **81.51** | **83.96** | 66.57 | **71.10** | **68.76** |

* marks the model trained on the fully clean dataset.

**Noisy-Label Learning.** As a baseline, the pre-trained teacher model achieves accuracy of 77.70% and 73.97% under the ResNet34 and VGGNet16 architectures, respectively. As shown in Table 2, MODUL+TAND consistently improves upon MODUL across all $\rho_1, \rho_2$ noise settings and even outperforms the teacher model under certain low-noise scenarios. The gains over MODUL, though modest in absolute percentage, are stable across noise regimes, underscoring the effectiveness of variance reduction. This indicates that TAND prevents the student from being misled by erratic gradients caused by teacher miscalibration or open-set contamination, thereby improving robustness.

Table 2: Accuracy comparison of different distillation methods under various label ratios.

| Architectures | Methods | $\rho_1 = 0.25$ | | | $\rho_1 = 0.50$ | | | $\rho_1 = 0.75$ | | |
|---|---|---|---|---|---|---|---|---|---|---|
| Teacher + Student | | $\rho_2 = 0.25$ | $\rho_2 = 0.50$ | $\rho_2 = 0.75$ | $\rho_2 = 0.25$ | $\rho_2 = 0.50$ | $\rho_2 = 0.75$ | $\rho_2 = 0.25$ | $\rho_2 = 0.50$ | $\rho_2 = 0.75$ |
| ResNet34 + ResNet18 | MODUL | 77.15 | 77.34 | 75.57 | 76.56 | 76.27 | 74.23 | 74.67 | 73.61 | 72.39 |
| | MODUL + TAND | **78.25** | **77.84** | **76.33** | **77.14** | **76.57** | **76.19** | **75.55** | **73.69** | **72.76** |
| VGGNet16 + VGGNet13 | MODUL | 73.65 | 73.16 | 72.44 | 73.54 | 72.45 | 71.59 | 71.84 | 70.25 | 69.03 |
| | MODUL + TAND | **74.51** | **74.13** | **73.77** | **74.13** | **73.35** | **72.35** | **72.26** | **71.08** | **69.32** |

**Transfer Gap.** Table 3 shows results under both same-architecture and cross-architecture distillation. IPWD+TAND achieves consistent improvements over IPWD, indicating that TAND enhances reliability even when teacher and student representations differ significantly. This demonstrates that decoupled, normalized supervision better bridges the mismatch between human-labeled and machine-labeled distributions.

Table 3: Accuracy comparison of different distillation methods with the same and different architecture styles.

| Methods | Same architecture style | | | | Different architecture style | | | |
|---|---|---|---|---|---|---|---|---|
| | WRN-40-2 WRN-16-2 | WRN-40-2 WRN-40-1 | resnet56 resnet20 | resnet32x4 resnet8x4 | ResNet50 MobileNetV2 | resnet32x4 ShuffleNetV1 | WRN-40-2 ShuffleNetV1 | vgg13 MobileNetV2 |
| Teacher | 75.60 | 75.60 | 72.41 | 79.42 | 79.33 | 79.42 | 75.6 | 73.69 |
| IPWD | 72.60 | 70.04 | 68.91 | 70.25 | 62.87 | 69.41 | 71.29 | 61.88 |
| TAND+IPWD | **73.50** | **70.91** | **69.79** | **71.50** | **64.24** | **70.52** | **72.76** | **63.75** |

## 4.3 ABLATION STUDIES

We report ablation studies on the Twitter dataset for the DS-NER task to evaluate the contribution of each component, as similar findings are consistently observed in noisy-label learning and transfer-gap settings. As shown in Table 4, removing either TCKD or NCKD leads to substantial performance degradation, underscoring the necessity of distinguishing between target and non-target signals. In addition, excluding NKD lowers both precision and recall, confirming its crucial role in stabilizing gradients through the normalization of non-target distributions. Overall, these results demonstrate that both NKD and TAD are indispensable for ensuring the robustness of TAND.

## 4.4 LOSS ANALYSIS OF NKD AND TAD

We analyze the loss behavior of TAD and NKD on the Twitter dataset for the DS-NSR task.

Table 4: Ablation study of TAND components.

| Ablations | P | R | F1 |
|---|---|---|---|
| w/o TCKD | 58.76 | 45.62 | 51.38 |
| w/o NCKD | 57.13 | 45.18 | 50.46 |
| w/o NKD | **58.64** | 45.18 | 51.04 |
| TAND | 57.96 | **47.68** | **52.32** |

**NKD.** Table 5 evaluates the effects of NKD's target and non-target losses. Both components contribute positively, and their combination achieves the highest F1, confirming that the synergy between target and non-target supervision is critical for robust distillation.

Table 5: Ablation study of NKD's target and non-target loss.

| Loss | F1-Score (%) | | | |
|---|---|---|---|---|
| Target | ✗ | ✓ | ✗ | ✓ |
| Non-target | ✗ | ✗ | ✓ | ✓ |
| | *51.86* | 51.89 | 52.27 | **52.32** |

**TAD.** Figure 2 analyzes sensitivity to $\alpha$ and $\beta$. Results show that proper balancing yields consistent gains, and TCKD is indispensable: removing it leads to unstable optimization. This confirms that explicit decoupling not only stabilizes learning but also offers tunable flexibility to emphasize either accuracy or calibration as needed.

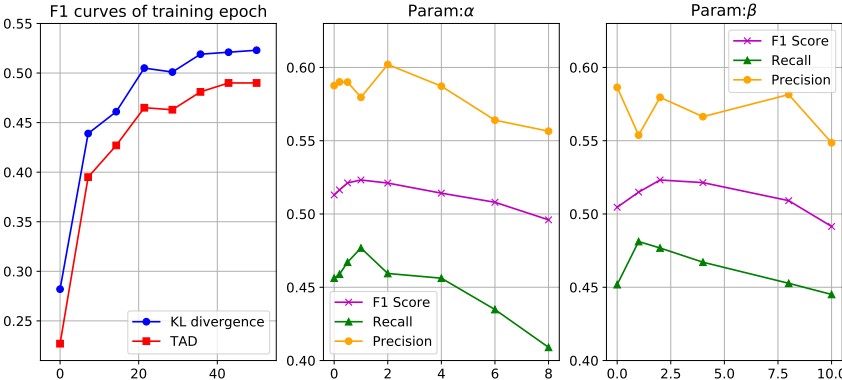

Figure 2: Parameter analysis of TAD.

Across DS-NER, noisy-label learning, and transfer gap, TAND consistently enhances baseline performance by mitigating gradient variance and eliminating structural pathologies of KL-based KD. Ablation and parameter studies confirm that both NKD and TAD are essential and complementary. Overall, TAND emerges as a general and robust distillation framework applicable across domains with weak supervision, noisy labels, or distributional mismatch.

## 5 CONCLUSION

*Knowledge Distillation* (*KD*) has become a fundamental paradigm for model compression, semi-supervised learning, and self-training. However, its standard KL-based objective inherently couples supervision on target and non-target classes, which renders optimization fragile in the presence of noise, miscalibration, or distributional shift. To overcome these limitations, we introduced *Target-Aware Normalized Distillation* (*TAND*), a principled framework that explicitly decouples and normalizes distillation signals. TAND integrates two complementary components: *Normalized KD* (*NKD*), which aligns normalized non-target distributions to mitigate mass mismatch, and *Target-Aware Distillation* (*TAD*), which assigns independent weights to target and non-target terms to eliminate magnitude coupling. This design reduces gradient variance, stabilizes training dynamics, and enables explicit control over supervision strength. Both theoretical analysis and extensive experiments on distantly supervised NER, noisy-label learning, and transfer-gap scenarios confirm that TAND consistently outperforms standard KL-based KD, offering a general and robust distillation objective applicable across diverse learning regimes.

ETHICS STATEMENT

This work makes use of publicly available datasets and models. No private or sensitive data is involved, and no harmful content is included. Therefore, we believe this paper does not raise any ethical concerns.

REPRODUCIBILITY STATEMENT

For our proposed models and existing compared algorithms, the implementation details are given in Appendix B, and the codes will be released upon the publication of this paper. For theoretical results, clear explanations of any assumptions and a complete proof of the claims are included in Appendix A.

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

## A PROOFS

### A.1 DECOMPOSITION OF KL DIVERGENCE

*Proof of Theorem 2.1.* By definition, the KL divergence between teacher and student distributions is

$$\mathrm{KL}(p^t \, \| \, p^s) = \sum_{c=1}^{C} p_c^t \log \frac{p_c^t}{p_c^s},$$

where $t$ denotes the ground-truth class. Splitting the sum into target and non-target terms gives

$$\mathrm{KL}(p^t \, \| \, p^s) = p_t^t \log \frac{p_t^t}{p_t^s} + \sum_{c \neq t} p_c^t \log \frac{p_c^t}{p_c^s}.$$

Rewriting the non-target part by factoring out the total non-target mass $(1 - p_t^t)$ yields

$$\sum_{c \neq t} p_c^t \log \frac{p_c^t}{p_c^s} = (1 - p_t^t) \sum_{c \neq t} \frac{p_c^t}{1 - p_t^t} \log \frac{p_c^t}{p_c^s}.$$

Now define the normalized non-target distributions

$$\rho_c^t = \frac{p_c^t}{1 - p_t^t}, \qquad \rho_c^s = \frac{p_c^s}{1 - p_t^s}, \qquad c \neq t.$$

Substituting these definitions gives

$$\sum_{c \neq t} \frac{p_c^t}{1 - p_t^t} \log \frac{p_c^t}{p_c^s} = \sum_{c \neq t} \rho_c^t \log \frac{\rho_c^t (1 - p_t^t)}{\rho_c^s (1 - p_t^s)}.$$

This expression separates into two terms:

$$\sum_{c \neq t} \rho_c^t \log \frac{\rho_c^t}{\rho_c^s} + \log \frac{1 - p_t^t}{1 - p_t^s} \sum_{c \neq t} \rho_c^t.$$

Since $\sum_{c \neq t} \rho_c^t = 1$, this simplifies to

$$\mathrm{KL}(\rho^t \, \| \, \rho^s) + \log \frac{1 - p_t^t}{1 - p_t^s}.$$

Putting everything together, we obtain

$$\begin{aligned}
\mathrm{KL}(p^t \, \| \, p^s) &= p_t^t \log \frac{p_t^t}{p_t^s} + (1 - p_t^t) \log \frac{1 - p_t^t}{1 - p_t^s} + (1 - p_t^t) \mathrm{KL}(\rho^t \, \| \, \rho^s) \\
&= \mathrm{KL}\big([p_t^t, 1 - p_t^t] \, \| \, [p_t^s, 1 - p_t^s]\big) + (1 - p_t^t) \mathrm{KL}(\rho^t \, \| \, \rho^s) \\
&= L_{\mathrm{TCKD}} + (1 - p_t^t) L_{\mathrm{NCKD}}.
\end{aligned}$$

This matches the decomposition claimed in Theorem 2.1. $\square$

### A.2 OPTIMIZATION STABILITY ANALYSIS OF TAND

To establish why TAND yields more stable optimization than standard KD, we first derive the exact gradient form of KD, then compare gradient norms under KD, TAD, and NKD, and finally show that TAND achieves provable variance reduction via a delta-method argument.

**Lemma A.1** (Gradient form of standard KD)**.** *Let $z^s \in \mathbb{R}^C$ be the student logits and $\tau > 0$ a temperature. Define $p^s = \mathrm{softmax}(z^s / \tau)$ and a fixed teacher distribution $p^t \in \Delta^{C-1}$. For the KD loss*

$$L_{\mathrm{KD}} = \mathrm{KL}\big(p^t \, \| \, p^s\big) = \sum_{c=1}^{C} \left[ p_c^t \cdot \log \left( \frac{p_c^t}{p_c^s} \right) \right],$$

*the gradient with respect to the student logits is*

$$\nabla_{z^s} L_{\mathrm{KD}} = \frac{1}{\tau} \cdot \left( p^s - p^t \right).$$

*In particular, the target vs. non-target gradient ratio is fixed by $(p^s - p^t)$ and is not a tunable hyperparameter under standard KD.*

*Proof of Lemma A.1.* We differentiate $L_{\mathrm{KD}}$ w.r.t. the student logits $z^s$. Since $p^t$ is constant, the loss can be written (up to an additive constant independent of $z^s$) as the cross-entropy

$$L_{\mathrm{KD}} = -\sum_{c=1}^{C} p_c^t \log p_c^s + \mathrm{const.}$$

Let $u = z^s/\tau$ so that $p^s = \mathrm{softmax}(u)$ with coordinates $p_c^s = \frac{e^{u_c}}{\sum_{k=1}^{C} e^{u_k}}$. We will first compute $\frac{\partial L_{\mathrm{KD}}}{\partial u}$ and then apply the chain rule $\frac{\partial u}{\partial z^s} = \frac{1}{\tau} I$.

*Step 1: gradient w.r.t. $u$.* The Jacobian of the softmax has the well-known form

$$\frac{\partial p_c^s}{\partial u_d} = p_c^s (\mathbf{1}\{c = d\} - p_d^s).$$

Using $\frac{\partial L_{\mathrm{KD}}}{\partial p_c^s} = -\frac{p_c^t}{p_c^s}$, the chain rule gives, for each $d \in \{1, \ldots, C\}$,

$$\frac{\partial L_{\mathrm{KD}}}{\partial u_d} = \sum_{c=1}^{C} \frac{\partial L_{\mathrm{KD}}}{\partial p_c^s} \frac{\partial p_c^s}{\partial u_d} = \sum_{c=1}^{C} \left( -\frac{p_c^t}{p_c^s} \right) p_c^s (\mathbf{1}\{c = d\} - p_d^s)$$

$$= \sum_{c=1}^{C} \left( -p_c^t \mathbf{1}\{c = d\} + p_c^t p_d^s \right) = -p_d^t + p_d^s \sum_{c=1}^{C} p_c^t$$

$$= -p_d^t + p_d^s \cdot 1 = p_d^s - p_d^t.$$

Thus, in vector form,

$$\nabla_u L_{\mathrm{KD}} = p^s - p^t.$$

*Step 2: chain rule back to $z^s$.* Since $u = z^s/\tau$, we have $\frac{\partial u}{\partial z^s} = \frac{1}{\tau} I$, hence

$$\nabla_{z^s} L_{\mathrm{KD}} = \left( \frac{\partial u}{\partial z^s} \right)^{\top} \nabla_u L_{\mathrm{KD}} = \frac{1}{\tau} \left( p^s - p^t \right).$$

This completes the proof. □

We now compare gradient magnitudes and smoothness.

**Lemma A.2** (Gradient norms). *For KD, $\|\nabla_{z^s} L_{\mathrm{KD}}\|_2 = \|p^s - p^t\|_2/\tau$ (Lemma A.1). For TAD,*

$$\left\| \nabla_{z^s} L_{\mathrm{TAD}} \right\|_2 \leq \frac{1}{\tau} \left( \alpha \left\| b^s - b^t \right\|_2 + \beta \left\| \rho^s - \rho^t \right\|_2 \right),$$

*For NKD, the non-target term uses $(\hat{p}^s, \hat{p}^t)$ with $\sum_{c \neq t} \hat{p}_c^s = 1$, which yields $\|J_{\hat{p} \leftarrow z}\|_{\mathrm{op}} \leq \|J_{p \leftarrow z}\|_{\mathrm{op}} \cdot \max\{1, m^{-1}\}$ where $m = 1 - p_t^s$, while its loss coefficient is a constant $\gamma$ that is independent of $m$ (no explicit $m^t$ multiplier as in standard KD).*

*Proof of Lemma A.2. KD.* By Lemma A.1, $\nabla_{z^s} L_{\mathrm{KD}} = \frac{1}{\tau}(p^s - p^t)$, hence $\|\nabla_{z^s} L_{\mathrm{KD}}\|_2 = \|p^s - p^t\|_2/\tau$.

*TAD: target term.* Recall $L_{\mathrm{TAD}} = \alpha \mathrm{KL}(b^t \| b^s) + \beta \mathrm{KL}(\rho^t \| \rho^s)$ with $b^s = [p_t^s, 1 - p_t^s]$ and $u = z^s/\tau$. Introduce the two-logit reparameterization

$$v = \begin{bmatrix} v_t \\ v_{\neg t} \end{bmatrix} := \begin{bmatrix} u_t \\ \log \sum_{k \neq t} e^{u_k} \end{bmatrix}, \qquad b^s = \mathrm{softmax}(v).$$

In these coordinates, the gradient of the two-class cross-entropy is *exact*: $\nabla_v \mathrm{CE}(b^t, b^s) = b^s - b^t$. By the chain rule,

$$\nabla_u \mathrm{KL}(b^t \| b^s) = \nabla_u \mathrm{CE}(b^t, b^s) = \left( \frac{\partial v}{\partial u} \right)^\top (b^s - b^t).$$

The Jacobian $\partial v / \partial u$ has operator norm at most 1: the first row selects the $t$-th coordinate, and the second is the gradient of $\log \sum_{k \neq t} e^{u_k}$, which is a probability vector on the non-target simplex (thus $\ell_2$-norm $\leq 1$). Therefore,

$$\|\nabla_u \mathrm{KL}(b^t \| b^s)\|_2 \leq \|b^s - b^t\|_2.$$

Finally, $\nabla_{z^s} = (1/\tau) \nabla_u$ gives

$$\left\| \nabla_{z^s} \alpha \, \mathrm{KL}(b^t \| b^s) \right\|_2 \leq \frac{\alpha}{\tau} \|b^s - b^t\|_2.$$

*TAD: non-target term.* Observe that $\rho^s$ is the softmax over the *non-target* logits:

$$\rho_c^s = \frac{p_c^s}{1 - p_t^s} = \frac{e^{u_c} / \sum_k e^{u_k}}{\sum_{j \neq t} e^{u_j} / \sum_k e^{u_k}} = \frac{e^{u_c}}{\sum_{j \neq t} e^{u_j}} = \mathrm{softmax}(u_{\neg t})_c, \quad c \neq t.$$

Consequently, if we view $u_{\neg t}$ as the parameter for $\rho^s$, then

$$\nabla_{u_{\neg t}} \mathrm{KL}(\rho^t \| \rho^s) = \rho^s - \rho^t \quad \Rightarrow \quad \nabla_u \mathrm{KL}(\rho^t \| \rho^s) = \left( \frac{\partial u_{\neg t}}{\partial u} \right)^\top (\rho^s - \rho^t),$$

where $\partial u_{\neg t} / \partial u$ simply selects the non-target coordinates (operator norm 1). Then

$$\|\nabla_u \mathrm{KL}(\rho^t \| \rho^s)\|_2 \leq \|\rho^s - \rho^t\|_2.$$

Scaling back to $z^s$ yields

$$\left\| \nabla_{z^s} \beta \, \mathrm{KL}(\rho^t \| \rho^s) \right\|_2 \leq \frac{\beta}{\tau} \|\rho^s - \rho^t\|_2.$$

Combining target and non-target parts by the triangle inequality proves the stated TAD bound.

*NKD: Lipschitz of the normalization map and mass-independence of the coefficient.* Recall

$$L_{\mathrm{NKD}} = -p_t^t \log p_t^s - \gamma \sum_{c \neq t} \hat{p}_c^t \log \hat{p}_c^s, \qquad \hat{p}_c^s = \frac{p_c^s}{\sum_{j \neq t} p_j^s} = \frac{p_c^s}{m}, \quad m = 1 - p_t^s.$$

Write $\hat{p}^s = T(p^s)$ with $T : \Delta^{C-1} \to \Delta^{C-2}, T(p)_{c \neq t} = p_c / (1 - p_t)$. A direct Jacobian computation for $T$ gives, for $c \neq t$ and $d \neq t$,

$$\frac{\partial \hat{p}_c^s}{\partial p_d^s} = \frac{1}{m} \left( \delta_{cd} - \hat{p}_d^s \right), \qquad \frac{\partial \hat{p}_c^s}{\partial p_t^s} = \frac{\hat{p}_c^s}{m}.$$

Hence, for any perturbation $h \in \mathbb{R}^C$,

$$\|J_{\hat{p} \leftarrow p} h\|_2 \leq \frac{1}{m} \left( \|h_{\neg t}\|_2 + \|\hat{p}^s\|_2 |h_t| \right) \leq \frac{1}{m} \|(h_{\neg t}, h_t)\|_2 \leq \max\{1, m^{-1}\} \|h\|_2,$$

which implies $\|J_{\hat{p} \leftarrow p}\|_{\mathrm{op}} \leq \max\{1, m^{-1}\}$. By the chain rule,

$$\|J_{\hat{p} \leftarrow z}\|_{\mathrm{op}} = \|J_{\hat{p} \leftarrow p} J_{p \leftarrow z}\|_{\mathrm{op}} \leq \|J_{\hat{p} \leftarrow p}\|_{\mathrm{op}} \|J_{p \leftarrow z}\|_{\mathrm{op}} \leq \|J_{p \leftarrow z}\|_{\mathrm{op}} \cdot \max\{1, m^{-1}\}.$$

Finally, note that in $L_{\mathrm{NKD}}$ the non-target term is multiplied by the *constant* $\gamma$ (chosen by the practitioner) and *not* by the teacher's non-target mass $m^t$. Thus, unlike standard KD where the non-target loss carries an explicit $m^t$ factor (via the decomposition $\mathrm{KL}(p^t \| p^s) = \mathrm{KL}(b^t \| b^s) + m^t \mathrm{KL}(\rho^t \| \rho^s)$), NKD's coefficient does not scale with $(m^t, m^s)$, preventing amplification when $m^t$ fluctuates. $\square$

KD has an exact gradient norm $\|p^s - p^t\|_2 / \tau$. TAD's gradient norm is bounded by a sum of target and non-target discrepancies scaled respectively by $\alpha/\tau$ and $(\beta/\tau)\|J_{\rho \leftarrow z}\|_{\mathrm{op}}$. For NKD, the normalization map contributes at most a factor $\max\{1, m^{-1}\}$ to the Jacobian, while the loss-level coefficient of the non-target term is a constant $\gamma$ independent of the non-target masses.

*Proof of Theorem 3.1.* Let $u = z^s/\tau$, $p^s = \text{softmax}(u)$ and decompose vectors into the target coordinate and the non-target block:

$$d := p^s - p^t = \begin{bmatrix} d_t \\ d_{\neg t} \end{bmatrix} = \begin{bmatrix} p_t^s - p_t^t \\ p_{\neg t}^s - m^t \rho^t \end{bmatrix}, \quad \text{where } \rho^t = \frac{p_{\neg t}^t}{m^t} \in \Delta^{C-2}.$$

By Lemma A.1, $\nabla_{z^s} L_{\text{KD}} = \frac{1}{\tau} d$ and hence $\|\nabla_{z^s} L_{\text{KD}}\|_2 = \frac{1}{\tau} \|d\|_2$.

Consider the (random, step-dependent) scalar function

$$g_{\text{KD}}(m) := \big\| [d_t,\ p_{\neg t}^s - m\,\rho^t] \big\|_2, \qquad \|\nabla_{z^s} L_{\text{KD}}\|_2 = \frac{1}{\tau}\, g_{\text{KD}}(m^t).$$

Treat all other quantities ($p_t^s$, $p_{\neg t}^s$, $\rho^t$) as locally fixed when differentiating w.r.t. $m$ (we return to their slow drift below). By standard differentiation of norms,

$$\frac{\mathrm{d}}{\mathrm{d}m}\, g_{\text{KD}}(m) = \frac{\langle [d_t,\ p_{\neg t}^s - m\rho^t],\ [0,\ -\rho^t] \rangle}{\big\| [d_t,\ p_{\neg t}^s - m\rho^t] \big\|_2} = \frac{-\langle p_{\neg t}^s - m\rho^t,\ \rho^t \rangle}{g_{\text{KD}}(m)}.$$

In particular, at a typical operating point $\bar{m} := \mathbb{E}[m^t]$ (or any fixed linearization point),

$$\big|\, g_{\text{KD}}'(\bar{m})\, \big| = \frac{\big|\langle p_{\neg t}^s - \bar{m}\rho^t,\ \rho^t \rangle\big|}{g_{\text{KD}}(\bar{m})} \geq \frac{c_\rho}{g_{\text{KD}}(\bar{m})} \quad \text{for some } c_\rho > 0,$$

whenever the non-target discrepancy $p_{\neg t}^s - \bar{m}\rho^t$ is not orthogonal to $\rho^t$ (a non-degenerate condition during training). By the delta method (first-order variance propagation),

$$\text{Var}\big[g_{\text{KD}}(m^t)\big] = \big(g_{\text{KD}}'(\bar{m})\big)^2 \text{Var}[m^t] + \text{(higher-order terms)}.$$

Consequently,

$$\text{Var}\big[\|\nabla_{z^s} L_{\text{KD}}\|_2\big] = \frac{1}{\tau^2}\, \text{Var}\big[g_{\text{KD}}(m^t)\big] = \frac{(g_{\text{KD}}'(\bar{m}))^2}{\tau^2}\, \text{Var}[m^t] + \text{h.o.t.}$$

For TAND, write its gradient norm schematically as

$$\|\nabla_{z^s} L_{\text{TAND}}\|_2 = \frac{1}{\tau}\, \big\| \alpha\, \Gamma_{\text{tar}}(b^s, b^t) + \beta\, \Gamma_{\text{cond}}(\rho^s, \rho^t) + \gamma\, \Gamma_{\text{norm}}(\hat{p}^s, \hat{p}^t) \big\|_2,$$

where each $\Gamma_\bullet$ denotes the corresponding (vector-valued) gradient contribution (see Lemma A.2). Crucially, none of these terms is multiplied at the *loss level* by $m^t$; the dependence on $m^t$ enters only indirectly through the *shape* $\rho^t$ (and $\hat{p}^t$) rather than via a coefficient. Linearizing again in $m$,

$$\text{Var}\big[\|\nabla_{z^s} L_{\text{TAND}}\|_2\big] = \frac{(g_{\text{TAND}}'(\bar{m}))^2}{\tau^2}\, \text{Var}[m^t] + \text{h.o.t.},$$

for some $g_{\text{TAND}}$ whose derivative $g_{\text{TAND}}'(\bar{m})$ is *controlled* by $(\alpha, \beta, \gamma)$ and the Jacobians from $m$ to $(\rho^t, \hat{p}^t)$. Under mild regularity, these Jacobians are bounded, and since there is no explicit $m^t$ multiplier, one obtains

$$\big|g_{\text{TAND}}'(\bar{m})\big| \leq C_\rho\, \beta\, \|J_{\rho \leftarrow m}\|_{\text{op}} + C_{\hat{p}}\, \gamma\, \|J_{\hat{p} \leftarrow m}\|_{\text{op}},$$

for constants $C_\rho, C_{\hat{p}}$ depending on local norms of $(\rho^s - \rho^t)$ and $(\hat{p}^s - \hat{p}^t)$. Comparing with $g_{\text{KD}}'(\bar{m})$, which has a *direct* nonzero contribution proportional to $\|\rho^t\|$ (no Jacobian attenuation), there exists

$$\kappa = \frac{(g_{\text{KD}}'(\bar{m}))^2 - (g_{\text{TAND}}'(\bar{m}))^2}{\tau^2} > 0$$

whenever $(\alpha, \beta, \gamma)$ are chosen in the usual ranges (e.g., $\beta, \gamma \leq 1$) and the non-degeneracy condition above holds. Subtracting the two delta-method expansions yields

$$\text{Var}\big[\|\nabla_{z^s} L_{\text{TAND}}\|_2\big] \leq \text{Var}\big[\|\nabla_{z^s} L_{\text{KD}}\|_2\big] - \kappa\, \text{Var}[m^t] + \text{(higher-order terms)}.$$

The "higher-order terms" collect curvature of $g_\bullet$ w.r.t. $m$, as well as slow co-variation of $(p^s, \rho^t)$ with $m$; these are second order in the local Taylor expansion and are empirically small when $m^t$ is the dominant source of variability. $\qquad\square$

# B COMPLEMENTARY EXPERIMENTS

## B.1 DS-NER

We perform a case study using specific instances from Table 6 to comprehend the benefits of the proposed TAND approach. We present the prediction results of a test sequence using BOND, SCDL, and TAND on the CoNLL03 dataset with label noise. BOND demonstrates the modest ability to generalize towards unseen mentions and alleviate some incomplete annotations. SCDL, through collaborative training, achieves better generalization for more accurate entity detection. However, these methods remain susceptible to label noise. For example, both BOND and SCDL are able to locate "British" and "American", but fail to precisely identify the entire entity context. In contrast, our TAND adeptly leverages soft labels to discern target categories through perceptual understanding, rather than only relying on distant labels (e.g., "British Airways", "American Airways").

Table 6: Case study: different colors denote distinct entity types.

| |
|---|
| **Distant Labels**: One focus was British Airways, which rebounded after talks between the [U.S. Transportation Department and]$_{ORG}$ British government may jeopardise its tie-up with American Airlines. |
| **Golden Labels**: One focus was [British Airways]$_{ORG}$, which rebounded after talks between the [U.S. Transportation Department]$_{ORG}$ and [British government]$_{ORG}$ may jeopardise its tie-up with [American Airlines]$_{ORG}$. |
| **BOND**: One focus was [British]$_{LOC}$ Airways, which rebounded after talks between the [U.S. Transportation Department and]$_{ORG}$ [British government]$_{ORG}$ may jeopardise its tie-up with [American]$_{LOC}$ Airlines. |
| **SCDL**: One focus was [British]$_{LOC}$ Airways, which rebounded after talks between the [U.S. Transportation Department]$_{ORG}$ and [British government]$_{ORG}$ may jeopardise its tie-up with [American]$_{LOC}$ Airlines. |
| **TAND**: One focus was [British Airways]$_{ORG}$, which rebounded after talks between the [U.S. Transportation Department]$_{ORG}$ and [British government]$_{ORG}$ may jeopardise its tie-up with [American Airlines]$_{ORG}$. |

## B.2 IMPLEMENTATION DETAILS

**DS-NER.** We reproduce SCDL Zhang et al. (2021) using its provided code and implement SCDL+TAND on top of it.

**Noisy-abel learning.** We reproduce the MODUL algorithm following the implementation in Tang et al. (2024), and extend it to implement MODUL+TAND.

**Transfer gap.** We implement IPWD based on the codes from Tian et al. (2019), as the original IPWD paper (Niu et al., 2022) did not provide codes. Building on this implementation, we further develop IPWD+TAND.

## THE USE OF LARGE LANGUAGE MODELS (LLM)

We use LLMs to assist with text polishing, as well as to search for and retrieve related work. All ideas and contributions, however, originate solely from the authors.

