# OpenReview forum: "Target-Aware Normalized Distillation: A Principled Framework for Robust Knowledge Transfer"
_ICLR.cc/2026/Conference — ICLR 2026 Conference Withdrawn Submission_

### Official Review · Reviewer_Ae2r · 2025-10-28

**Soundness:** 1
**Presentation:** 2
**Contribution:** 2
**Rating:** 2
**Confidence:** 4

**Summary:**

This paper highlights that conventional KL-based knowledge distillation (KD) suffers from structural issues, particularly magnitude coupling and mass mismatch, when the data contain noise or the teacher model is miscalibrated, leading to undesirable coupling between target and non-target components. To address these issues, the authors propose Target-Aware Normalized Distillation (TAND), a framework that combines Normalized KD (NKD), which removes non-target mass mismatch through normalization, and Target-Aware Distillation (TAD), which explicitly decouples target and non-target terms with independent weighting. The resulting objective theoretically reduces gradient variance and improves optimization stability. Empirical results on DS-NER, noisy-label learning, and transfer gap settings show consistent gains over KL-based KD, especially in noisy or miscalibrated scenarios.

**Strengths:**

* The paper provides a reasonable analysis of two structural limitations, magnitude coupling and mass mismatch, in standard KL-based KD, offering an interpretable view of why performance can degrade under uncertainty or noise.
* The proposed TAND framework introduces normalization and explicit weighting to address these issues within the loss formulation, representing a straightforward and simple modification.
* Experiments show that TAND yields more stable or slightly improved results under noisy or miscalibrated conditions, suggesting robustness without introducing significant complexity.

**Weaknesses:**

* The empirical validation lacks methodological rigor. Although the proposed method introduces three additional hyperparameters $(\alpha, \beta, \gamma)$, the paper provides no guidelines or validation procedure for setting them. Figure 2 only illustrates sensitivity but does not explain how these parameters should be determined in practice. Moreover, there is no discussion on the effect of the temperature parameters, which can naturally mitigate teacher miscalibration. The warm-up procedure using the KL divergence term in the noisy-label and transfer gap experiments also lacks theoretical or empirical justification. As a result, the reported improvements are difficult to interpret as evidence of effectiveness under well-designed experimental conditions.
* The paper focuses on noisy or miscalibrated settings, where TAND shows clear benefits. However, in realistic scenarios, it is often difficult to know in advance whether such conditions exist. Therefore, it remains unclear whether TAND can maintain or at least not degrade performance under benign settings with clean data and well-calibrated teachers. A more thorough analysis of this aspect would strengthen the practical relevance of the method.
* The methodological clarity of the proposed components is somewhat limited. While NKD and TAD are claimed to address mass mismatch and magnitude coupling separately, both effectively include the same non-target term $\sum_{k \neq t} \rho^t_k \log \rho^s_k$, making their functional distinction unclear. Consequently, the combined loss could be reduced to a single equivalent term without introducing both $\beta$ and $\gamma$, which raises questions about the necessity of separating NKD and TAD. This overlap obscures the individual contribution of each component and makes the formulation harder to interpret.

**Questions:**

* The paper argues that the coupling between target and non-target components is a structural problem in KL-based KD. Could you clarify under what conditions this coupling becomes detrimental? Given that softmax normalization inherently couples class probabilities, how can we distinguish harmful coupling from the natural dependency induced by softmax itself?
* In Equation (3), when the teacher assigns high confidence to the target class, the non-target mass becomes very small. Could you elaborate on why this reduction in the influence of the NCKD term should be viewed as problematic? Does the proposed decoupling mainly help when the teacher is miscalibrated or uncertain, and how does it behave when the teacher is already well-calibrated and confident?

---

### Official Review · Reviewer_636T · 2025-10-29

**Soundness:** 3
**Presentation:** 3
**Contribution:** 3
**Rating:** 4
**Confidence:** 3

**Summary:**

The authors propose TAND for knowledge distillation. Standard KD algorithms couples the loss for both the target and non-target terms in the probability distribution. This paper, argues that such coupling is bad because it causes instability in training potentially due to calibration errors. The authors then propose to calculate loss for target and non-target terms separately.

**Strengths:**

1. The paper is well-written and the diagram in Figure 1 summarizes the approach very well.
2. The  authors show theoretical analyses to motivate their approach.

**Weaknesses:**

1. Improvements could be a result of hyper-parameter tuning. The authors propose three losses, balanced with three additional hyper-parameters. It's possible the improvements come from excessive hyper-parameter sweeping rather than the method itself.

2. The experiments are a bit weak. The datasets used are from 2003 and 2015. It would be great if the authors can also try some more modern models and datasets.

3. In-depth analyses. I appreciate the authors for showing the ablation study for each of the losses. However, I think the paper can benefit from more in-depth analyses regarding the intuition of the approach. For example, the authors claim issues like magnitude coupling and mass mismatch. I think it'd be great if the authors can demonstrate these empirically.

**Questions:**

N/A

---

### Official Review · Reviewer_3ctW · 2025-10-31

**Soundness:** 2
**Presentation:** 3
**Contribution:** 1
**Rating:** 2
**Confidence:** 4

**Summary:**

This paper presents a technique that decouples the learning dynamics of target and non-target knowledge. While similar approaches have been explored in previous works, such as Decoupled Knowledge Distillation (CVPR 2022), the authors do not explicitly reference or discuss the relationship with this prior research. The primary contribution appears to be demonstrating that this method can reduce optimization variance. The authors validate their approach through comprehensive experiments in various noisy-label learning scenarios.

**Strengths:**

- The paper is well-written and easy to follow.
- It addresses an important research problem: how to perform more effective knowledge distillation from teachers with noisy outputs.

**Weaknesses:**

- **Limited Novelty.**
This work appears to share considerable similarities with a prior study [1]. It might be beneficial if the authors could explicitly address or discuss the connections between their approach and this existing work, as such clarification would help readers better appreciate the novel contributions of the current study.

- **Lack of Comparative Baselines.**
Numerous knowledge distillation techniques have been proposed in recent years (e.g., [2, 3]). Including comparisons with some of these established methods would strengthen the experimental evaluation and provide clearer positioning of the proposed approach.

- **Clarification of Mathematical Notation.**
Some mathematical notations could be further clarified for better readability. For instance, the notation $p_t^t$, where the subscript and superscript both use the same letter 't' (seemingly referring to both the teacher model and target class), may lead to potential ambiguity. Consider using more distinct symbols to avoid confusion.

[1] Decoupled knowledge distillation. CVPR 2022

[2] Logit standardization in knowledge distillation. CVPR 2024

[3] Knowledge distillation based on transformed teacher matching. ICLR 2024

**Questions:**

See Weaknesses.

---

### Official Review · Reviewer_cB8b · 2025-11-01

**Soundness:** 3
**Presentation:** 3
**Contribution:** 2
**Rating:** 4
**Confidence:** 3

**Summary:**

The paper decomposes the standard KD objective into a target-versus-rest term and a normalized non-target term (Theorem 2.1). On top of this, it proposes two losses: Normalized KD (NKD), which matches the target probability and the normalized non-target distribution, and Target-Aware Distillation (TAD), which assigns explicit weights to the target and non-target terms. The unified objective TAND is defined as NKD plus TAD. The paper claims that TAND removes coupling between terms and reduces the variance of gradient norms relative to vanilla KD, with a variance-reduction statement given as Theorem 3.1.

**Strengths:**

Clean algebraic decomposition of KL that makes explicit the role of the non-target mass. The statement is correct and easy to verify, and it helps motivate a decoupled treatment.

The proposed objectives are simple to state, and the paper gives a consistent narrative about magnitude coupling and mass mismatch, together with design desiderata for decoupling and explicit control.

**Weaknesses:**

1. Theorem 2.1 is a direct identity that follows from partitioning probability mass into target and non-target subsets. Beyond that identity, there is no theorem that shows the proposed objectives are optimal or even preferable with respect to a principled risk. There is no characterization of the solution that TAND targets, no KKT analysis, and no link to a Bregman projection on the probability simplex. As a result, the main mathematical novelty remains modest.

2.TAD sets (L_{\text{TAD}}=\alpha \cdot \text{TCKD}+\beta\cdot\text{NCKD}). This can be seen as replacing the random factor (m_t=1-p_{tt}) with a constant (\beta). However, there is no proof that this corresponds to minimizing any proper scoring rule, any calibrated surrogate, or any Bregman divergence. A theorem that TAD minimizes a well-defined risk, or that it is the solution of a constrained projection with tunable duals ((\alpha,\beta)), would add needed substance.

3. Theorem 3.1 states that the variance of the gradient norm under TAND is less than that under KD, minus (\kappa\mathrm{Var}[p_{tt}]), plus higher order terms, for some (\kappa>0) that depends on (\alpha,\beta,\gamma). The result hinges on what randomness the variance is over, what independence or boundedness assumptions are made, and how (\kappa) is bounded away from zero. None of these are made explicit. The “higher-order terms” are also not quantified. Without a clean set of assumptions and an explicit expression or lower bound for (\kappa), the inequality can become non-falsifiable. The authors should specify the probability space, the role of mini-batch sampling, and the smoothness constants used in the proof.

**Questions:**

1. Can you prove that LNKD is minimized at (p_s=p_t) and characterize the optimizer, including uniqueness, when (\gamma>0)? If not, what weaker property do you guarantee, such as consistency of the target mass and the normalized non-target distribution at a stationary point?

2. What are the exact assumptions behind Theorem 3.1? Please define the probability space, the source of randomness, any independence assumptions, and provide an explicit lower bound on (\kappa(\alpha,\beta,\gamma)) that does not collapse to zero when the weights are small.

3. Can you state conditions on (\alpha,\beta,\gamma) that ensure well-posedness, bounded gradients, and nontrivial training signals on a trimmed simplex (p_t\in[\delta,1-\delta])?

4. Please specify a numerically stable definition of (\rho_t) and (\rho_s) near the boundary, for example with (\varepsilon)-smoothing, and show continuity of the gradients as (p_{tt}\to 1) or (p_{st}\to 1).

5. Can you connect TAD to a principled divergence or constrained projection, for example by showing that it minimizes a weighted Bregman divergence where (\alpha,\beta) arise as dual variables of simple linear constraints?

---

### Note · Authors · 2025-11-17

I have read and agree with the venue's withdrawal policy on behalf of myself and my co-authors.